# Prevalence of Vitamin D Deficiency Among Adults in Kazakhstan: A Systematic Review and Meta-Analysis

**DOI:** 10.3390/medicina60122043

**Published:** 2024-12-11

**Authors:** Indira Karibayeva, Galiya Bilibayeva, Aya Yerzhanova, Roza Alekesheva, Assiya Iglikova, Makhigul Maxudova, Neilya Ussebayeva

**Affiliations:** 1Department of Health Policy and Community Health, Jiann-Ping Hsu College of Public Health, Georgia Southern University, Statesboro, GA 30460, USA; 2Department of Nursing, Faculty of Medicine and Public Health, Al-Farabi Kazakh National University, Almaty 050038, Kazakhstan; bilibayeva@kaznu.kz (G.B.); yerzhanova.aya@kaznu.kz (A.Y.); alekesheva@kaznu.kz (R.A.); maksudova63@gmail.com (M.M.); usebaeva@mail.ru (N.U.)

**Keywords:** vitamin D, vitamin D deficiency, adults, prevalence, Kazakhstan, systematic review, meta-analysis

## Abstract

*Background and Objectives*: Despite frequent references to the high prevalence of vitamin D deficiency in Kazakhstan, a comprehensive synthesis of existing research on this issue among adults is lacking. This systematic review and meta-analysis aim to address this gap by determining the mean prevalence of vitamin D deficiency among adults in Kazakhstan. A secondary objective is to evaluate whether the prevalence differs between healthy adults and those with chronic conditions. *Materials and Methods*: A systematic search was conducted in PubMed, Scopus, ScienceDirect, Web of Science, and Google Scholar by two independent researchers using the keywords: “vitamin D” AND “Kazakhstan”, following the Preferred Reporting Items for Systematic Reviews and Meta-Analyses guidelines. Studies were included if they reported the prevalence of vitamin D in adults in Kazakhstan, defined as a blood level of 25-hydroxyvitamin D 25(OH)D below 20 ng/mL. *Results*: Seven studies were included in this review, encompassing 3616 individuals, of whom 2239 had vitamin D deficiency. Using a random-effects model, the pooled mean prevalence of vitamin D deficiency among adults with chronic conditions was 60% (95% CI, 38–79%), with high heterogeneity. Similarly, the analysis of five studies involving healthy adults revealed a pooled mean prevalence of 55% (95% CI, 38–70%), also with high heterogeneity. Overall, the pooled mean prevalence of vitamin D deficiency in the adult population was 57% (95% CI, 45–69%). *Conclusions*: This analysis indicates a high prevalence of vitamin D deficiency among adults in Kazakhstan, with 57% of the population affected. Addressing this issue requires a multifaceted approach, including policy reforms that consider the impact of time zone changes on sunlight exposure and the active involvement of nurse practitioners in preventive strategies.

## 1. Introduction

Vitamin D is integral to the regulation of calcium and phosphate metabolism and is involved in melatonin synthesis and mitochondrial regulation, thereby playing a pivotal role in maintaining bone integrity, mineral balance, and immune system function [1,2]. Primarily synthesized in the skin through the ultraviolet-B radiation-induced conversion of 7-dehydrocholesterol, vitamin D can also be obtained from dietary sources, notably fatty fish [3]. However, factors such as limited sun exposure, sedentary indoor lifestyles, inadequate dietary intake, higher latitudes, obesity-related sequestration of vitamin D metabolites in adipose tissue, and certain medical conditions impairing absorption contribute to the widespread prevalence of vitamin D deficiency globally [4,5].

Kazakhstan, a landlocked nation spanning approximately 2,724,900 square kilometers, is frequently noted for its high prevalence of vitamin D deficiency [6,7,8]. Geographically, the country extends from the 40th parallel north at its southernmost point to the 56th parallel north at its northernmost reaches, placing it well above the Equator. This northern latitude results in prolonged winters with reduced ultraviolet B exposure, significantly limiting endogenous vitamin D synthesis. Furthermore, traditional dietary patterns in this landlocked region are deficient in key vitamin D sources such as fish [9]. Modern indoor lifestyles further reduce sun exposure, even during warmer seasons, compounding the risk of deficiency.

Despite frequent references to the high prevalence of vitamin D deficiency in Kazakhstan, a comprehensive synthesis of existing research on this issue among adults is lacking. This systematic review and meta-analysis aim to address this gap by estimating the mean prevalence of vitamin D deficiency among adults in Kazakhstan. A secondary objective is to evaluate whether the prevalence varies between healthy adults and those with chronic conditions.

## 2. Materials and Methods

The study protocol is registered with the National Institute for Health Research’s PROSPERO International Prospective Register of Systematic Reviews [10] (ID: CRD42024610447).

### 2.1. Search Strategy

An initial search of the PROSPERO database to identify registrations of comparable studies revealed one study protocol that assessed the mean vitamin D levels among the Kazakhstani population [11]. Since the objective of the present study was to evaluate the mean prevalence of vitamin D deficiency among adult population in Kazakhstan, the authors proceeded with registering the current study protocol in the PROSPERO database. Following this, a systematic search was conducted across five electronic databases: PubMed, Scopus, ScienceDirect, Web of Science, and Google Scholar. The search commenced on 2 July 2024, and concluded on 1 November 2024. The search strategy utilized the keywords: “vitamin D” AND “Kazakhstan”. No restrictions were placed on publication date; however, the results were limited to English- and Russian-language publications and studies conducted on humans. Where applicable, filters were applied to include only research articles and exclude other publication types.

### 2.2. Eligibility Criteria

The types of studies to be included were determined using the following eligibility criteria: The inclusion criteria were as follows: 1. Studies reporting the number of adults in the population in Kazakhstan with vitamin D deficiency, defined as a blood level of 25-hydroxyvitamin D 25(OH)D below 20 ng/mL. 2. Observational studies including cohort studies, cross-sectional studies, and case–control studies. 3. Studies on the adult population in Kazakhstan. 4. Studies published in English or Russian languages. The exclusion criteria were as follows: 1. Studies reporting the mean value of 25-hydroxyvitamin D in serum or lacking required information. 2. Studies with duplicative results. 3. Case reports, reviews, editorials, and conference abstracts. 4. Studies published in languages other than English or Russian.

### 2.3. Selection of Studies and Data Extraction

The literature review and synthesis were conducted according with the Preferred Reporting Items for Systematic Reviews and Meta-Analyses (PRISMA) guidelines [12]. Two authors independently screened the titles and abstracts of the search results, following duplicate removal, to assess their relevance. Full texts of studies that met the initial criteria were retrieved and evaluated against the pre-defined inclusion and exclusion criteria. Data extracted from eligible studies included the first author’s last name, year of publication, region where the study was conducted, description of the included population (e.g., disease name, if applicable, or healthy), sample size, number of male participants, mean age, mean serum 25(OH)D levels and number of adults with confirmed vitamin D deficiency. Two authors independently performed data extraction from the selected studies. Any discrepancies were resolved through consultation with a third author to ensure consensus among all three authors responsible for the study selection and data extraction process.

### 2.4. Risk of Bias (Quality) Assessment

The studies included in this review were evaluated for risk of bias (quality) using the Newcastle–Ottawa Scale (NOS) for case–control studies and its adapted version for cross-sectional studies, as recommended by the Cochrane Non-Randomized Studies Methods Working Group [13]. The NOS for case–control studies assesses each study based on eight criteria divided into three main categories: selection of study groups (four questions), comparability of these groups (one question), and exposure (three questions). Each criterion is assigned up to one point, while the comparability criterion can receive a maximum of two points. The total score ranges from 0 to 9, with higher scores indicating better study quality. The adapted NOS for cross-sectional studies evaluates each study across six criteria divided into three main categories: selection (three questions), comparability (one question), and outcome (two questions). Each criterion is assigned up to one point, and the comparability criterion can receive a maximum of two points, yielding a total score ranging from 0 to 7, with higher scores reflecting better study quality. Quality assessments were conducted independently by two authors (IK and GB) after agreeing on the assessment procedures. A third author (AY) calculated the inter-rater agreement between the two assessors. In this review, studies scoring seven points or higher were classified as high-quality, while those scoring four points or less were deemed low-quality.

We conducted a self-assessment of the potential risk of bias (quality) assessment in our systematic review using A Measurement Tool to Assess Systematic Reviews-2 (AMSTAR-2), which evaluates the methodological quality of systematic reviews, including those with randomized and non-randomized clinical studies [14]. AMSTAR-2 comprises 16 domains that assess various aspects such as protocol registration, thoroughness of the literature search, study selection, data extraction, risk of bias evaluation, heterogeneity, and reporting of results. Each domain is rated to identify potential biases or methodological weaknesses, culminating in an overall quality rating of low, moderate, or high.

### 2.5. Statistical Strategy for Data Synthesis

We utilized RStudio software (version 4.3.2) with the ‘meta’ and ‘metafor’ packages to calculate the pooled mean prevalence of vitamin D deficiency among adults in Kazakhstan, using a random-effects model for the meta-analysis [15,16]. Forest plots were generated to display the pooled estimates. A meta-regression analysis was conducted with age and population size as covariates. To explore sources of heterogeneity, we performed an influence analysis. The generalizability of the study findings was assessed through a publication bias evaluation, which included visual inspection of a funnel plot and statistical analysis using Egger’s test. Additionally, to further investigate sources of heterogeneity, we conducted a subgroup analysis comparing healthy adults to those with chronic conditions [16].

## 3. Results

### 3.1. Included Study Characteristics

The initial database search yielded 226 articles. After removing 61 duplicates, 165 unique articles remained for screening. Of these, 33 were retrieved for full-text review following the exclusion of 132 non-relevant titles. Upon further assessment, seven articles met the inclusion criteria. Ten studies lacked data on vitamin D deficiency, six focused on children [17,18,19,20,21,22], five were excluded for other reasons, and four were reviews. A PRISMA flow diagram detailing the study selection process is presented in Figure 1 [12].

The included studies were published between 2015 and 2024. Two studies were conducted across multiple regions, two in Karaganda, two in Astana, and one in Almaty city. Among these, two studies did not include healthy controls, one was conducted among a healthy population only, and the remaining studies involved both patients with chronic diseases and healthy controls. Collectively, the seven studies assessed a total of 3616 individuals, identifying vitamin D deficiency in 2239 participants. Detailed information on the included articles is provided in Table 1.

### 3.2. Prevalence of Vitamin D Deficiency

Six studies, covering seven groups, reported on the prevalence of vitamin D deficiency among patients with various health conditions. Using a random-effects model, the pooled mean prevalence of vitamin D deficiency in adults with chronic conditions was 60% (95% CI, 38–79%), with high heterogeneity. In contrast, analysis of five studies on healthy adults using a random-effects model yielded a pooled mean prevalence of 55% (95% CI, 38–70%), also showing high heterogeneity. The overall pooled mean prevalence of vitamin D deficiency in the adult population was 57% (95% CI, 45–69%) (Figure 2).

### 3.3. Meta-Regression Analysis

A meta-regression analysis revealed a negative impact of age on the pooled mean prevalence of vitamin D deficiency. The analysis indicates that as age increases, the prevalence of vitamin D deficiency also rises, although the finding was not statistically significant (*p* > 0.05). (Figure 3).

### 3.4. Heterogeneity Assessment

An influence analysis was conducted to identify the studies with the greatest impact on the pooled estimate. The pooled mean estimate was primarily influenced by study #2, by Nugmanova 2015 (b), which included patients with a detectable HIV viral load [23] (Figure 4a). The leave-one-out analysis confirmed these results, showing that when the Nugmanova 2015 (b) study was excluded, the pooled estimate reached its highest value at 53% (Figure 4b).

### 3.5. Publication Bias Assessment

Upon visual inspection of the funnel plot, no asymmetry was evident (Figure 5). The absence of publication bias was further confirmed by non-significant results from Egger’s test for publication bias (*p* > 0.05).

### 3.6. Risk of Bias (Quality) Assessment Results

All included case–control studies had a NOS score of 7 or higher out of 8, and all cross-sectional studies had a NOS score of 6 out of 7, indicating the high quality and a low risk of bias of the included studies, as presented in Table 2.

The AMSTAR-2 evaluation, detailed in the Appendix A, indicates that this systematic review has a moderate level of methodological quality.

## 4. Discussion

Our systematic review and meta-analysis reveals a significant prevalence of vitamin D deficiency among adults in Kazakhstan, with an overall rate of 57% (95% CI, 45–69%). Notably, the prevalence is slightly lower among healthy adults (55%, 95% CI, 38–70%) and higher among those with chronic conditions (60%, 95% CI, 38–79%). Additionally, mean serum 25(OH)D levels were below 20 ng/mL in four out of five studies that reported them, particularly among adults with chronic conditions. These findings align with the existing literature, which indicates high rates of vitamin D deficiency in northern-hemisphere countries, especially those with limited fish consumption, underscoring a pressing public health concern.

Addressing this widespread deficiency requires a comprehensive, multi-tiered strategy at both the policy and primary healthcare levels. This approach aligns with the World Health Organization’s framework for the global monitoring of non-communicable diseases [28], given the limited fish consumption and sunlight exposure in the region.

Undoubtedly, policy-level interventions should focus on addressing social determinants of health and ensuring an accessible, high-quality health system capable of delivering essential preventive services. Kazakhstan has made significant progress in increasing healthcare system financing through the introduction of the Compulsory Social Health Insurance System in January 2020, which aims to improve healthcare service delivery [29]. Nevertheless, policy interventions should also encourage outdoor physical activities to enhance sun exposure and decrease the risk of cardiovascular diseases and diabetes, the leading causes of mortality among adults in Kazakhstan [30]. A policy change in March 2024, which unified the country under a single time zone (UTC + 5) [31], may have inadvertently reduced sunlight exposure and negatively impacted public health. This shift resulted in earlier winter sunsets, around 4 PM, limiting daylight hours for outdoor activities, especially for those engaging in such activities for work. Given Kazakhstan’s vast expanse of approximately 2700 km from west to east, naturally spanning time zones from UTC + 3 to UTC + 6, the adoption of a single time zone may not align with the country’s geographical and solar realities [31].

The second implication of our study is the need for educational initiatives targeting the general population. An Endocrine Society Clinical Practice Guideline on vitamin D does not recommend empirical vitamin D supplementation beyond the current Dietary Reference Intake or advocate routine screening for 25(OH)D levels in the general population [32]. Empirical supplementation of vitamin D can be achieved through a combination of fortified foods [33] and daily food supplements, rather than through intermittent high doses to reduce the risk of disease in healthy adults under the age of 75 [32]. Thus, primary healthcare providers and nurse practitioners, in particular, play a crucial role in implementing these preventive strategies at the community level [34]. The ongoing interactions of nurse practitioners with both healthy individuals and patients with chronic conditions position them uniquely to educate on the importance of vitamin D, its dietary sources, and lifestyle modifications to improve vitamin D levels. Moreover, a study by Gregor and Sebach showed the high effectiveness of a nurse-practitioner-led vitamin D intervention program in a primary care setting [35]. In Kazakhstan, as in other post-Soviet countries, the practice of dispanserization—regular monitoring of patients with chronic diseases—facilitates frequent interactions between nurse practitioners and patients [36]. This framework enables nurse practitioners to deliver personalized guidance and raise awareness about effective, evidence-based interventions for vitamin D supplementation.

### Limitations

This study has certain limitations. The lack of region-specific data prevents an analysis of potential geographical variations in vitamin D deficiency across Kazakhstan. Additionally, we did not assess the presence of metabolic conditions, such as obesity, diabetes, and hypertension, in the patient group. This omission is significant, as these conditions are known to influence vitamin D metabolism and status [37]. Furthermore, the absence of information regarding the timing of vitamin D measurements is a constraint, as vitamin D levels are known to fluctuate seasonally. We also did not evaluate the extent of sun exposure among participants, a factor that directly affects vitamin D synthesis [38]. Finally, subjectivity in study selection, data extraction, and interpretation could introduce reviewer bias. To mitigate this, the authors employed recommended strategies, such as utilizing a standardized protocol, conducting assessments in duplicate, and resolving discrepancies through consensus and third-party adjudication [14].

## 5. Conclusions

Our analysis reveals a high prevalence of vitamin D deficiency among Kazakhstan’s adult population, with 57% affected. The deficiency rate remains concerning among healthy adults at 55%, and even higher in adults with chronic conditions at 60%. Addressing this issue requires a multifaceted approach, which includes policy reforms that take into account the effects of time zone changes on sunlight exposure, as well as the active involvement of nurse practitioners in implementing preventive educational strategies.

### Future Research Directions

To build upon our findings and address the limitations of the current study, future research should focus on the following: (1) investigating the impact of metabolic disorders on serum vitamin D levels in the general population; (2) assessing variations in serum vitamin D levels relative to sunlight exposure, considering factors such as seasonality and regional differences; (3) conducting meta-analyses to evaluate mean vitamin D levels across different demographics, including both adults and children, to inform targeted interventions. By pursuing these research avenues, we can develop a more comprehensive understanding of vitamin D deficiency in Kazakhstan, ultimately leading to more effective public health strategies and policies.

## Figures and Tables

**Figure 1 medicina-60-02043-f001:**
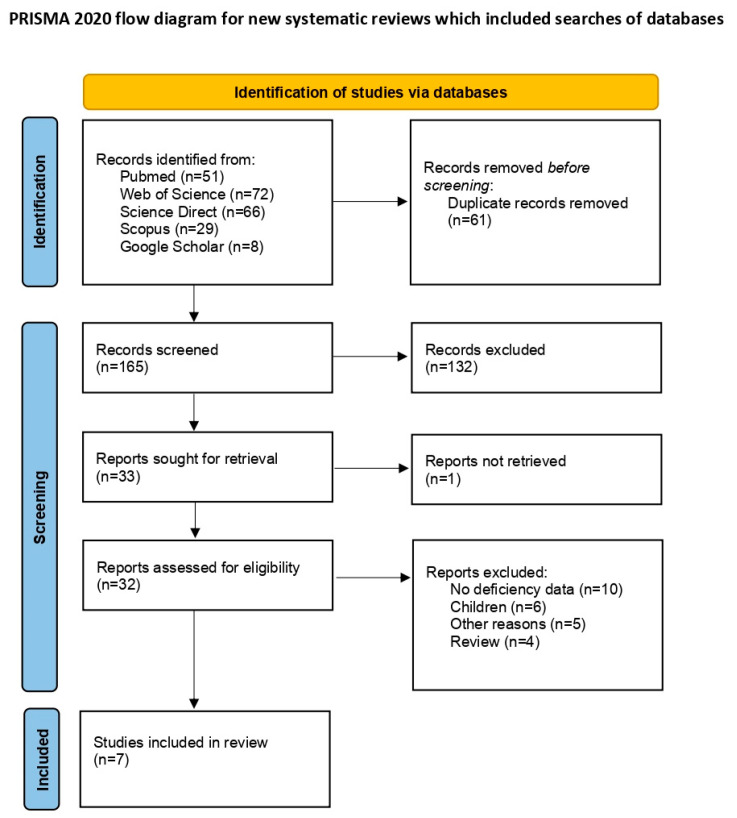
PRISMA flow diagram of study selection process.

**Figure 2 medicina-60-02043-f002:**
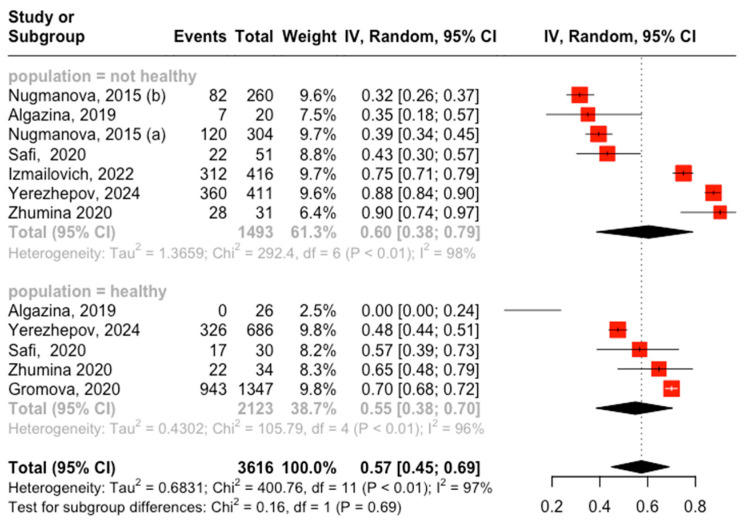
Forest plot of vitamin D deficiency among adults in Kazakhstan. Abbreviations: CI—confidence interval; HIV—Human Immunodeficiency Virus; Nugmanova, 2015: (a) HIV viral load undetectable; Nugmanova, 2015 (b): HIV viral load detectable.

**Figure 3 medicina-60-02043-f003:**
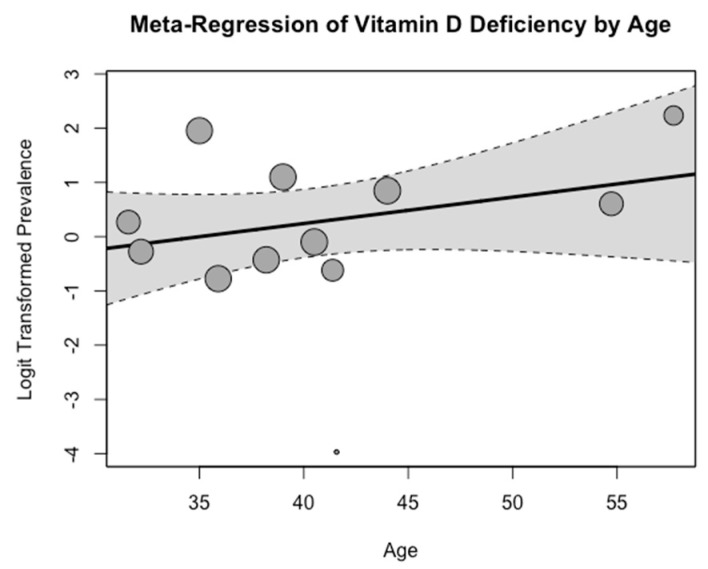
Meta-regression analysis of vitamin D deficiency by age.

**Figure 4 medicina-60-02043-f004:**
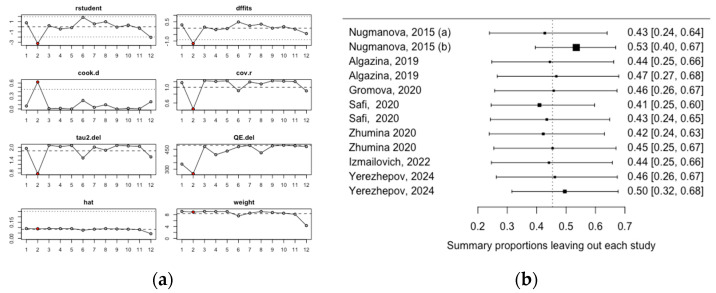
Influence analysis of vitamin D deficiency: (**a**) influence analysis; (**b**) leave-one-out analysis. Abbreviations: Nugmanova, 2015: (**a**) HIV viral load undetectable; Nugmanova, 2015 (**b**): HIV viral load detectable.

**Figure 5 medicina-60-02043-f005:**
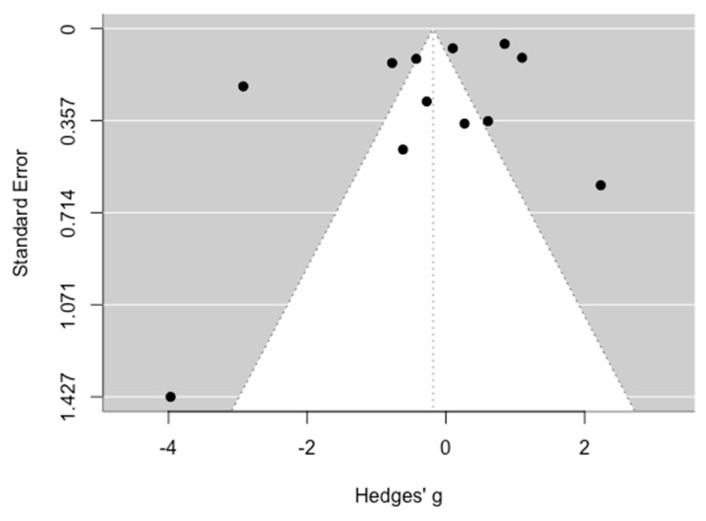
Publication bias assessment.

**Table 1 medicina-60-02043-t001:** Description of the included studies.

First Author, Year	Region/City	Study Design	Disease Name/Healthy	Sample	Male	Age (Mean ± SD)	25(OH)D (ng/mL) (Mean ± SD)	Numberwith VDD
Nugmanova, 2015 [23]	Almaty	Case–control	HIV undet. (a) HIV det. (b)	304 (a) 260 (b)	168 (a) 143 (b)	38.2 ± 9.0 (a) 35.9 ± 8.7 (b)	25.90 ± 13.20 (a) 26.40 ± 11.40 (b)	120 (a) 82 (b)
Algazina, 2019 [24]	Astana	Case–control	Psoriasis Healthy	20 (NH) 26 (H)	10 (NH) 13 (H)	41.38 ± 4.2 (NH) 41.57 ± 5.67 (H)	NA	7 (NH) 0 (H)
Gromova, 2020 [7]	Multiple	Cross-sectional	Healthy	1347	528	44 ± 14	NA	943
Safi, 2020 [25]	Astana	Case–control	PCOS Healthy	51 (NH) 30 (H)	0 0	18–44	16.25 17.52	22 (NH) 17 (H)
Zhumina, 2020 [26]	Karaganda	Case–control	Leukemia Healthy	31 (NH) 34 (H)	14 (NH) 17 (H)	57.71 ± 13.7 (NH) 54.73 ± 15.1 (H)	10.85 ± 7.00 (NH) 21.61 ± 7.78 (H)	28 (NH) 22 (H)
Izmailovich, 2022 [27]	Karaganda	Cross-sectional	Allergic Rhinitis	416	149	39 ± 8	16.10 ± 6.90	312
Yerezhepov, 2024 [8]	Multiple	Case–control	Tuberculosis Healthy	411 (NH) 686 (H)	224 (NH) 303 (H)	35 ± 13.1 (NH) 40.5 ± 13.9 (H)	12.90 ± 3.80 (NH) 24.80 ± 3.10 (H)	360 (NH) 326 (H)

Abbreviations: 25(OH)D—25-hydroxyvitamin D; H—healthy; HIV—Human Immunodeficiency Virus; NA—not available; NH—not healthy; PCOS—polycystic ovarian syndrome; SD—standard deviation.

**Table 2 medicina-60-02043-t002:** Newcastle–Ottawa risk of bias (quality) assessment results.

Study	Selection	Comparability	Exposure	Total
Case–control studies				
Nugmanova, 2015 [23]	4	1	3	8
Algazina, 2019 [24]	3	1	3	7
Safi, 2020 [25]	4	1	3	8
Zhumina, 2020 [26]	4	1	2	7
Yerezhepov, 2024 [8]	4	1	3	8
Cross-sectional studies				
Gromova, 2020 [7]	3	1	2	6
Izmailovich, 2022 [27]	3	1	2	6

## Data Availability

The raw data supporting the conclusions of this article will be made available by the authors on request.

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
