# Peer review of "Prevalence of Vitamin D Deficiency Among Adults in Kazakhstan: A Systematic Review and Meta-Analysis"

_medicina, 2024, doi:10.3390/medicina60122043_

Round 1

Reviewer 1 Report

Comments and Suggestions for Authors

Comments on the Quality of English Language

Revise english

Author Response

Dear Reviewer,

Thank you sincerely for dedicating your time and effort to enhance the quality and value of our manuscript for the readers.

Reviewer 1

The Editor Nov 25,2024
Medicina
Thank you very much for asking me to review following article.
Prevalence of Vitamin D Deficiency Among Adults in Kazakhstan: A Systematic Review and Meta-Analysis, by Indira Karibayeva et al. for consideration of publication.

  1. Abstract. Please give number of subjects studied along with number of studies; 3,616 with 2239 having deficiency.

Authors reply: The authors acknowledge the reviewer’s thoughtful comments and agree with the reviewer. The number of participants was added to the abstract in line 32 – 33: Seven studies were included in this review, encompassing 3,616 individuals, of whom 2,239 had vitamin D deficiency.

  1. Introduction
    Please also mention that Vitamin D is involved in melatonin synthsis as well as mitochondrial dysfunction which may predispose circadian misalighment leading to NCDs

Minich DM, Henning M, Darley C, Fahoum M, Schuler CB, Frame J. Is Melatonin the "Next

Vitamin D"?: A Review of Emerging Science, Clinical Uses, Safety, and Dietary Supplements.

Nutrients. 2022 Sep 22;14(19):3934.

Authors reply: The authors acknowledge the reviewer’s thoughtful comments and agree with the reviewer. The following changes have been made in line 48 – 50, and a reference was added: Vitamin D is integral to the regulation of calcium and phosphate metabolism and is involved in melatonin synthesis and mitochondrial regulation, thereby playing a pivotal role in maintaining bone integrity, mineral balance, and immune system function [1,2].

  1. Minich, D.M.; Henning, M.; Darley, C.; Fahoum, M.; Schuler, C.B.; Frame, J. Is Melatonin the “Next Vitamin D”?: A Review of Emerging Science, Clinical Uses, Safety, and Dietary Supplements. Nutrients 2022, Vol. 14, Page 3934 2022, 14, 3934–3944, doi:10.3390/NU14193934.

  1. Subjects and Methods may be Synthesis of Data

Exclusion criteria.Page 4 at base: Mention the number of studies in each category of

exclusion (n= )?

Authors reply: The authors acknowledge the reviewer’s thoughtful comments and agree with the reviewer. Number of excluded studies are provided in line 154 – 156, and in Figure 1: Ten studies lacked data on vitamin D deficiency, six focused on children [17–22], five were excluded for other reasons, and four were reviews.

  1. 4. Page 3. 1.5. Strategy for data synthesis

1.5 Statistical Strategy for data Synthesis is better than above subtitle.

Authors reply: The authors acknowledge the reviewer’s insightful comments and agree with the reviewer. We have revised the subtitle as follows in line 139: Statistical strategy for data synthesis

  1. Results
    The seven studies assessed a total of 3,616 individuals, identifying vitamin D deficiency in 2,239 participants.
    Did you look for presence of obesity, diabetes, hypertension among selected populations and so called healthy subjects, in which vit D deficiency was examined, if not the mention under limitations.
    Did you find out how many subjects have no opportunity for sun exposure?
    Table 1 should give demographic data on age, sex, body weight,BMI, mean BP, % of Obesity, hypertension, diabetes, CAD, stroke which are known to have vit D deficiency. It would be more useful to public health if you also give mean levels of Vit D in different studies and in the total number of subjects.

Authors’ reply: The authors acknowledge the reviewer’s thoughtful comments and agree with the reviewer. We did not assess the metabolic conditions of the patients, nor did we assess the opportunity for sun exposure. Thus, we added those raised points to the limitation of our study in line 263 – 269. Additionally, we did not assess the presence of metabolic conditions, such as obesity, diabetes, and hypertension, in the patient group. This omission is significant, as these conditions are known to influence vitamin D metabolism and status [37]. Furthermore, the absence of information regarding the timing of vitamin D measurements is a constraint, as vitamin D levels are known to fluctuate seasonally. We also did not evaluate the extent of sun exposure among participants, a factor that directly affects vitamin D synthesis [38].

However, we did collect the data on the mean values of vitamin D, and are currently working on the second manuscript regarding the mean vitamin D values in the population. The study protocol could be assessed on PROSPERO with the following reference ID: CRD42024598871.

Table 1 has also been updated to include the mean Vitamin D values from those studies that have provided them.

First author, year

Region/City

Study design

Disease name/healthy

Sample

Male

Age

(mean ± SD)

25(OH)D level (mean ± SD)

# with VDD

Nugmanova, 2015 [23]

Almaty

Case-control

HIV undet.(a) HIV det. (b)

304 (a)

260 (b)

168 (a)

143 (b)

38.2±9.0 (a) 35.9±8.7 (b)

25.90±      13.20 (a)

26.40±11.40 (b)

120 (a)

 82 (b)

Algazina, 2019 [24]

Astana

Case-control

Psoriasis

Healthy

20 (NH)

26 (H)

10 (NH)  13 (H)

41.38±4.2 (NH)  41.57±5.67 (H)

NA

7 (NH)

0 (H)

Gromova, 2020 [7]

Multiple

Cross-sectional

Healthy

1347

528

44 ±â€¯14

NA

943

Safi, 2020 [25]

Astana

Case-control

PCOS

Healthy

51 (NH)

30 (H)

0

0

18-44

16.25

17.52

22 (NH)

17 (H)

Zhumina, 2020 [26]

Karaganda

Case-control

Leukemia

Healthy

31 (NH)

34 (H)

14 (NH)

 17 (H)

57.71±13.7(NH)  54.73±15.1 (H)

10.85±7.00 (NH)

21.61±7.78 (H)

28 (NH) 

22 (H)

Izmailovich, 2022 [27]

Karaganda

Cross-sectional

Allergic Rhinitis

416

149

39±8

16.10±6.90

312

Yerezhepov, 2024 [8]

Multiple

Case-control

Tuberculosis

 Healthy

411 (NH) 686 (H)

224 (NH)

303 (H)

35±13.1 (NH) 40.5±13.9 (H)

12.90±3.80(NH)

24.80±3.10 (H)

360 (NH)

326 (H)

  1. Discussion.
    Mean levels of Vit D concentrations reported in various studies should be discussed along with confounders such food/fish intake and sun light exposure.

Authors’ reply: The authors acknowledge the reviewer’s thoughtful comments and agree with the reviewer. We have revised the discussion in the lines 218-220 and 227 to address the mean Vitamin D values, and limited fish consumption and sunlight exposure in the region:

Additionally, mean serum 25(OH)D levels were below 20 ng/ml in four out of five studies that reported them, particularly among adults with chronic conditions.

Addressing this widespread deficiency necessitates a comprehensive, multi-tiered strategy at the policy level and primary healthcare level, which is in line with the World Health Organization’s framework for global monitoring of non-communicable diseases [28], given the limited fish consumption and sunlight exposure in the region.

  1. References. Please give references published in 2023,2024

Authors’ reply: The authors acknowledge the reviewer’s thoughtful comments and agree with the reviewer. We have added and highlighted three additional recent references to the article #2, 36, and 37.

Reviewer 2 Report

Comments and Suggestions for Authors

I was glad to review the authors' work regarding this fascinating manuscript on the Prevalence of Vitamin D Deficiency Among Adults in Kazakhstan.

The manuscript is well-written and the topic is very interesting. I can consider acceptance for publication of the paper after minor changes.

1)     In the "quality of studies" section, you indicated the use of the Newcastle-Ottawa Scale (NOS), specifically designed for case-control studies; from this, it can be inferred that all the included studies are exclusively “case-control.” However, in the section where you describe the characteristics of the results, there is no mention of the types of studies found. I suggest adding information about the types of included studies to ensure consistency with what is written in the methods and to support the choice of the selected quality assessment tool (lines 148-153).

2)     Table 1: Regarding the previous comment, I would suggest adding a column with the study type; please also check the table font, as it appears different from the one used in the article's text.

3)     Figure 3: The title of the figure seems incomplete as it ends with "the": I assume it should be completed as "Vitamin D Deficiency by Age." Since the figure's title is included in the main text, I recommend removing it from the body of the figure.

4)     Figure 4: The reference letters for the boxes should be in lowercase as per the journal's guidelines; please correct this.

5)     Conclusions: The conclusions are clear and summarize the key points of the study. However, in the last sentence, there is a reference to the possible involvement of specialized nurses in preventive strategies. It would be advisable to briefly explain who they are, as this appears to be a professional specialization. Additionally, if the intention was to highlight the involvement of nurses in the implementation of preventive strategies, it would be helpful to mention which strategies could aid in improving vitamin deficiency. This would provide useful insights for developing dedicated preventive programs.

6)     References: The formatting of the references does not appear to comply with the journal's guidelines. For instance, article citations require the following format:

“Author 1, A.B.; Author 2, C.D. Title of the article. Abbreviated Journal Name YearVolume, page range.”

Please review and adjust if necessary.

Author Response

Reviewer 2:

I was glad to review the authors' work regarding this fascinating manuscript on the Prevalence of Vitamin D Deficiency Among Adults in Kazakhstan.

The manuscript is well-written and the topic is very interesting. I can consider acceptance for publication of the paper after minor changes.

  • In the "quality of studies" section, you indicated the use of the Newcastle-Ottawa Scale (NOS), specifically designed for case-control studies; from this, it can be inferred that all the included studies are exclusively “case-control.” However, in the section where you describe the characteristics of the results, there is no mention of the types of studies found. I suggest adding information about the types of included studies to ensure consistency with what is written in the methods and to support the choice of the selected quality assessment tool (lines 148-153).

Authors reply: The authors acknowledge the reviewer’s thoughtful comments and agree with the reviewer. We have revised the description of the NOS, and clarified that the NOS for case-control studies was used along with the its adapted version for cross-sectional studies in line 115 – 116: The studies included in this review were evaluated for risk of bias (quality) using the Newcastle-Ottawa Scale (NOS) for case-control studies and its adapted version for cross-sectional studies, as recommended by the Cochrane Non-Randomized Studies Methods Working Group [13].

We have also added the description of the adapted NOS for cross-sectional studies in line 122 – 126: The adapted NOS for cross-sectional studies evaluates each study across six criteria divided into three main categories: Selection (three questions), Comparability (one question), and Outcome (two questions). Each criterion is assigned up to one point, and the comparability criterion can receive a maximum of two points, yielding a total score ranging from 0 to 7, with higher scores reflecting better study quality.

Finally, the Table 2 (Results Section, page 7) has also been updated to show those changes:

Table 2. New-Castle Ottawa risk of bias (quality) assessment results

Study

Selection

Comparability

Exposure

Total

Case-control studies

Nugmanova, 2015

4

1

3

8

Algazina, 2019

3

1

3

7

Safi, 2020

4

1

3

8

Zhumina, 2020

4

1

2

7

Yerezhepov, 2024

4

1

3

8

Cross-sectional studies

Gromova, 2020

3

1

2

6

Izmailovich, 2022

3

1

2

6

  • Table 1: Regarding the previous comment, I would suggest adding a column with the study type; please also check the table font, as it appears different from the one used in the article's text.

Authors reply: The authors acknowledge the reviewer’s thoughtful comments and agree with the reviewer. Table 1 has been updated to include the study design

First author, year

Region/City

Study design

Disease name/healthy

Sample

Male

Age

(mean ± SD)

25(OH)D (ng/ml) (mean ± SD)

# with VDD

Nugmanova, 2015 [23]

Almaty

Case-control

HIV undet.(a) HIV det. (b)

304 (a)

260 (b)

168 (a)

143 (b)

38.2±9.0 (a) 35.9±8.7 (b)

25.90±      13.20 (a)

26.40±11.40 (b)

120 (a)

 82 (b)

Algazina, 2019 [24]

Astana

Case-control

Psoriasis

Healthy

20 (NH)

26 (H)

10 (NH)  13 (H)

41.38±4.2 (NH)  41.57±5.67 (H)

NA

7 (NH)

0 (H)

Gromova, 2020 [7]

Multiple

Cross-sectional

Healthy

1347

528

44 ±â€¯14

NA

943

Safi, 2020 [25]

Astana

Case-control

PCOS

Healthy

51 (NH)

30 (H)

0

0

18-44

16.25

17.52

22 (NH)

17 (H)

Zhumina, 2020 [26]

Karaganda

Case-control

Leukemia

Healthy

31 (NH)

34 (H)

14 (NH)

 17 (H)

57.71±13.7(NH)  54.73±15.1 (H)

10.85±7.00 (NH)

21.61±7.78 (H)

28 (NH) 

22 (H)

Izmailovich, 2022 [27]

Karaganda

Cross-sectional

Allergic Rhinitis

416

149

39±8

16.10±6.90

312

Yerezhepov, 2024 [8]

Multiple

Case-control

Tuberculosis

 Healthy

411 (NH) 686 (H)

224 (NH)

303 (H)

35±13.1 (NH) 40.5±13.9 (H)

12.90±3.80(NH)

24.80±3.10 (H)

360 (NH)

326 (H)

  • Figure 3: The title of the figure seems incomplete as it ends with "the": I assume it should be completed as "Vitamin D Deficiency by Age." Since the figure's title is included in the main text, I recommend removing it from the body of the figure.

Authors reply: The authors acknowledge the reviewer’s thoughtful comments and agree with the reviewer. Figure 3 title has been revised to: Figure 3. Meta-regression analysis of the Vitamin D deficiency by age

  • Figure 4: The reference letters for the boxes should be in lowercase as per the journal's guidelines; please correct this.

Authors reply: The authors acknowledge the reviewer’s thoughtful comments and agree with the reviewer. Figure 4 and its title were corrected.

  • Conclusions:The conclusions are clear and summarize the key points of the study. However, in the last sentence, there is a reference to the possible involvement of specialized nurses in preventive strategies. It would be advisable to briefly explain who they are, as this appears to be a professional specialization. Additionally, if the intention was to highlight the involvement of nurses in the implementation of preventive strategies, it would be helpful to mention which strategies could aid in improving vitamin deficiency. This would provide useful insights for developing dedicated preventive programs.

Authors reply: The authors acknowledge the reviewer’s thoughtful comments and agree with the reviewer. The intention was to highlight the role of nurses in educational interventions described in line 243 – 260. We have refined the conclusion to clarify our point as follows in line 280: Addressing this issue requires a multifaceted approach, which includes policy reforms that take into account the effects of time zone changes on sunlight exposure, as well as the active involvement of nurse practitioners in implementing preventive educational strategies.

  • References: The formatting of the references does not appear to comply with the journal's guidelines. For instance, article citations require the following format:

“Author 1, A.B.; Author 2, C.D. Title of the article. Abbreviated Journal Name Year, Volume, page range.”

Authors reply: The authors acknowledge the reviewer’s thoughtful comments and agree with the reviewer. References have been revised to match the suggested journal guidelines.
